# Combining Direct PCR Technology and Capillary Electrophoresis for an Easy-to-Operate and Highly Sensitive Infectious Disease Detection System for Shrimp

**DOI:** 10.3390/life12020276

**Published:** 2022-02-13

**Authors:** Hung-Yun Lin, Shao-Chieh Yen, Shou-Kuan Tsai, Fan Shen, John Han-You Lin, Han-Jia Lin

**Affiliations:** 1Center of Excellence for the Oceans, National Taiwan Ocean University, Keelung 20224, Taiwan; hungyun59@gmail.com; 2Department of Bioscience and Biotechnology, National Taiwan Ocean University, Keelung 20224, Taiwan; Max.Yen@bioptic.com.tw; 3BiOptic Inc., New Taipei City 23141, Taiwan; eric.tsai@bioptic.com.tw; 4Giant Bio Technology Inc., New Taipei City 22101, Taiwan; fanshen@giantbiotech.com; 5School of Veterinary Medicine, National Taiwan University, Taipei, 10617, Taiwan

**Keywords:** direct PCR, capillary electrophoresis, aquaculture, white spot syndrome virus, acute hepatopancreatic necrosis disease, *Enterocytozoon hepatopenaei*

## Abstract

Infectious diseases are considered the greatest threat to the modern high-density shrimp aquaculture industry. Specificity, rapidity, and sensitivity of molecular diagnostic methods for the detection of asymptomatic infected shrimp allows preventive measures to be taken before disease outbreaks. Routine molecular detection of pathogens in infected shrimp can be made easier with the use of a direct polymerase chain reaction (PCR). In this study, four direct PCR reagent brands were tested, and results showed that the detection signal of direct PCR in hepatopancreatic tissue was more severely affected. In addition, portable capillary electrophoresis was applied to improve sensitivity and specificity, resulting in a pathogen detection limit of 25 copies/PCR-reaction. Juvenile shrimp from five different aquaculture ponds were tested for white spot syndrome virus infection, and the results were consistent with the Organization for Animal Health’s certified standard method. Furthermore, this methodology could be used to examine single post larvae shrimp. The overall detection time was reduced by more than 58.2%. Therefore, the combination of direct PCR and capillary electrophoresis for on-site examination is valuable and has potential as a suitable tool for diagnostic, epidemiological, and pathological studies of shrimp aquaculture.

## 1. Introduction

As a globally traded animal providing agri-food with high quality protein, the global market for shrimp aquaculture was valued at USD 28 billion in 2018 [1,2]. Modern high-density aquaculture coupled with globalization has accelerated disease transmission [2,3]. Major shrimp infectious disease outbreaks on a global scale occur every few years, including white spot disease (WSD), acute hepatopancreatic necrosis disease (AHPND), and hepatopancreatic microsporidiosis (HPM), which have become prevalent in recent years [4]. According to recent studies, these diseases have caused an annual global economic loss of USD 7.8 billion [5,6,7].

As in the ongoing COVID-19 pandemic, until effective drugs are invented, non-pharmaceutical interventions (NPIs) are the most effective way to control the spread of infectious diseases [8]. In the shrimp aquaculture industry, quarantine measures for management and transfer are the most important means of NPI, and routine disease detection is an important foundation for the effective implementation of NPIs [9,10,11]. Effective molecular detection techniques for shrimp infectious diseases have been developed [12,13]. The World Organization for Animal Health (OIE) has standardized assays for major shrimp-borne diseases, most of which use nested PCR or real-time PCR technology [14]. Although these standard methods have high sensitivity and specificity for pathogen detection, the operational procedures are complicated, so they must be performed in a well-equipped laboratory. As a result, both the time and money needed for detection remain high, which limits the application of routine molecular detection and on-site examination of infectious diseases in aquaculture.

To make molecular diagnostics more convenient, new detection methods have been developed in recent years. Examples include recombinase polymerase amplification, loop-mediated isothermal amplification, and isothermal PCR technology, combined with the use of portable equipment to shorten the PCR detection time [15,16,17,18,19,20,21,22]. Another example is capillary electrophoresis (CE), which reduces post-PCR analysis time [23]. CE has been applied to various high-throughput nucleic acid analysis applications, including genotyping and pathogen detection, due to its high sensitivity, superior accuracy, speed, and automation [23,24,25]. The recent development of lightweight and portable CE machines makes this powerful tool an excellent apparatus for on-site nucleic acid detection [26]. However, regardless of the detection method, a nucleic acid purification step is required before PCR, which is inconvenient for on-site operation and can easily result in sample contamination during pathogen detection in aquaculture.

Direct PCR technology can omit the nucleic acid purification step and directly analyze shrimp tissue samples [27]. However, the PCR reaction may have interference from PCR inhibitors from tissues, resulting in reduced feasibility of use for diagnostics [28]. In this work, we carefully compared different direct PCR systems and found the best system for shrimp tissue analysis. By using portable CE to perform post-PCR analysis, the detection sensitivity, specificity, and efficiency was further improved [24]. After comparison with an OIE-certified system for disease detection of white shrimp in aquaculture ponds from different regions, we verified that this technology platform (i.e., direct PCR-based CE) is suitable for on-site disease surveillance in aquaculture.

## 2. Materials and Methods

### 2.1. Materials

Agarose, TAE buffer, and loading dye were provided by Ten Giga Bio for free (New Taipei, Taiwan). HealthView nucleic acid stain and nucleic acid standards were obtained from Genomics (New Taipei, Taiwan) and GeneDireX, Inc. (Taoyuan City, Taiwan), respectively. Milli-Q ultrapure water (18.2 MΩ·cm; EMD Millipore, Billerica, MA, USA) was used in all experiments.

### 2.2. PCR kits and Equipments 

Direct PCR kits used in this study were purchased from four different biotech companies: DirectGO (Ten Giga Bio, New Taipei, Taiwan), Quick Genotyping Kit (BioTools Co. Ltd., New Taipei, Taiwan), Terra PCR Direct Polymerase (Takara Bio Inc., Tokyo, Japan), and Phire Tissue Direct PCR (Thermo Fisher Scientific, Hudson, NH, USA). A *Qsep*_1_ portable capillary electrophoresis and an S2 cartridge CE column were purchased from BiOptic (New Taipei, Taiwan).

### 2.3. Source and Tissue Sampling of Shrimp

Juvenile shrimp (ca. 3 cm in length; 1 ± 0.4 g in body weight) and post larvae (ca. 0.5 cm in length; 3 ± 0.7 mg in body weight) of *Litopenaeus vannamei* were obtained from aquaculture ponds in five different regions of Taiwan: Toucheng, Taimali, Taimu, Linbian, and Jiadong. The on-site-collected shrimp were euthanized by immersion in ice for more than 15 min according to the guidelines of the American Veterinary Medical Association (AVMA). The shrimp were then sent to the laboratory immediately after freezing and stored at −80 °C. Before direct PCR, the frozen shrimp were left at room temperature for 5 min, followed by whole-body weighing, organ dissection, and tissue extraction. The organs (i.e., muscles, pereiopods, pleopods, gills, hemolymph, hepatopancreas, stomach, and midgut) of the juvenile shrimp were collected using dissection tools before being weighed and stored at −80 °C.

### 2.4. Direct PCR for the Detection of Shrimp Disease

The specific nucleic acid primers used in this study were referenced from previous reports [29,30,31], and detailed information is listed in Table 1, including the VP28 envelope protein gene of white spot syndrome virus (WSSV), the *Photorhabdus* insect-related (Pir) toxin gene in a plasmid of *Vibrio parahaemolyticus*, and the spore wall protein gene of *Enterocytozoon hepatopenaei*. Standard plasmids containing these pathogen-related genes were obtained from Omics Biotechnology Co., Ltd. (New Taipei, Taiwan) for whole gene synthesis and were cloned into the pUC57 plasmid. 

Tissue extraction was performed using the lysis buffer included in each direct PCR kit (from Ten Giga Bio, BioTools Co, Ltd., Takara Bio Inc., and Thermo Fisher Scientific). According to the manufacturer’s instructions, 3 mg of as-prepared tissue was added to 90 μL lysis buffer (30 μL mg^−1^) and heated at 95 °C for 15 min. After thermal extraction, the direct PCR reactions were initiated by adding 25 μL of sample, 400 nM specific primer pairs, and 2X Master Mix (from Ten Giga Bio, BioTools Co, Ltd., Takara Bio Inc., and Thermo Fisher Scientific). The reactions were carried out on a portable PCR thermal cycler (*Qamp*_mini_, BiOptic). The thermal profile was as follows: 10 min preheating at 95 °C; an amplification process with a denaturation step at 95 °C for 20 s; an annealing step at 55/64/58 °C for 20 s (depending on the specific primer); an extension step at 72 °C for 20 s with a total of 40 cycles; 5 min at 72 °C; and, finally, holding at 14 °C. PCR products were analyzed by portable capillary electrophoresis or gel electrophoresis on a 2% agarose gel after mixing with loading dye and nucleic acid stain. 

### 2.5. WSSV Detection in Juvenile Shrimp Using the IQ2000^TM^ WSSV Detection System

The dodecyltrimethylammonium bromide (DTAB)/cetyltrimethylammonium bromide (CTAB)-based nucleic acid extraction and nested PCR reaction of juvenile shrimp tissue samples followed the OIE recommendations [14]. Shrimp muscle tissue (20 mg) was added to 600 μL of DTAB, homogenized using a disposable grinder, and heated at 75 °C for 5 min. After cooling to room temperature, 700 μL of chloroform was added, and high-speed centrifugation (12,000× *g*, 25 °C, 5 min) was performed. Subsequently, 200 μL of the upper aqueous phase solution was mixed with 100 μL of CTAB solution and allowed to react at 75 °C for 5 min. The suspension was removed by high-speed centrifugation (12,000× *g*, 25 °C, 5 min), and the pellet was dissolved in 150 μL of dilution buffer and heated at 75 °C for 5 min. Finally, the pellet was dissolved in ultrapure water after washing with 95% and 75% ethanol.

### 2.6. Equations for Evaluating the Diagnostic Performance of Direct PCR-Based Capillary Electrophoresis

The diagnostic performance of the DirectGO kit was evaluated using three parameters: sensitivity, specificity, and agreement. The true positives (TP) or true negatives (TN) indicated whether the detection results of DirectGO were consistent with the IQ2000^TM^ WSSV detection system. In contrast, false positives (FP) and false negatives (FN) indicated whether the detection results of DirectGO were different from those of the IQ2000^TM^ WSSV detection system. The sensitivity was calculated as [TP/(TP + FN)] × 100%, the formula for specificity was [TN/(TN + FP)] × 100%, and the equation of agreement was [(TP + TN)/Total Sample] × 100%.

### 2.7. Ethics Statement 

All animal protocols in this study were reviewed and approved by the Institutional Animal Care and Use Committee, College of Life Sciences, National Taiwan Ocean University (IACUC Approval No. 102025).

## 3. Results

### 3.1. Disease Detection in Shrimp Tissue Using Direct PCR

To evaluate the feasibility of direct PCR with shrimp tissue samples (ca. 3 mg), we first collected the pereiopod, pleopod, gill, hemolymph, hepatopancreas, stomach, and midgut tissues from healthy juvenile Pacific whiteleg shrimp (*Litopenaeus vannamei*; ca. 3 cm in length and 1 ± 0.2 g in weight), followed by a simple thermal extraction process with a lysis buffer (30 μL mg^−1^). We spiked 250 copies/PCR-reaction of the WSSV viral gene fragment into each sample and then performed PCR analysis. The results showed that, under the same template concentration, the target gene fragments were successfully amplified in all sampled tissues (Figure 1). However, in the midgut, stomach, and hepatopancreas, the signal was decreased, with detection limits of 25, 100, and 100 copies/PCR-reaction, respectively (Figure 2A).

### 3.2. Increased Detection Limits Using Capillary Electrophoresis Analysis

Considering the disadvantages of analyzing PCR products with agarose gel-based electrophoresis, which include a higher detection limit (~5 ng DNA), longer analysis time, and more complex manual manipulation, we attempted to replace it with a portable capillary electrophoresis (CE) analyzer that offers better detection sensitivity and ease of operation. The results showed that, for the same sample, the sensitivity of the CE platform was more than four times higher than that of agarose gel-based electrophoresis (Figure 2B), and the entire analysis process took only 3 min.

Although CE has high sensitivity, according to previous reports, the formulation of PCR reagent kits by different brands may affect the analysis, especially the salt composition and concentration of the reaction buffer. Therefore, we chose four brands of direct PCR reagents available on the market (Thermo Fisher Scientific, Takara Bio Inc., BioTools Co, Ltd., and Ten Giga Bio). After the addition of the WSSV viral gene fragments, each PCR was performed, followed by CE analysis. The results showed that analytical sensitivity was affected by the type of PCR reagent brand. DirectGO of Ten Giga Bio exhibited the strongest signal, and the maximum signal difference was more than three times higher than the others (Figure 3A). The PCR product Terra PCR Direct Polymerase Mix (Takara Bio Inc.) even had a shift in size of the signal, which may be due to interference caused by the dye contained in the pre-mix kit. Next, we used the same infusion to perform direct PCR on the hepatopancreatic tissue to determine whether nucleic acid amplification efficiency was affected by interference from shrimp tissue extracts. Experiments were performed according to the operating conditions recommended by each manufacturer, with 250 copies of WSSV DNA spiked in each sample. A similar result showed that the highest peak was from the DirectGO (Ten Giga Bio) reaction product (Figure 3B).

The hepatopancreas is the main infection site for many shrimp pathogens, such as EHP and *Vibrio*. Therefore, we spiked the target DNA of EHP or *Vibrio* to the crude extract of hepatopancreatic tissue and analyzed it using a direct PCR-based CE platform. The results showed a specific and strong peak in each sample (Figure 4), indicating that either EHP or AHPND can be analyzed directly from hepatopancreatic tissue via a simple thermal extraction coupled with this technique.

### 3.3. Disease Detection in Adult Shrimp Tissue Using Direct PCR-Based Capillary Electrophoresis

A total of 20 tested whiteleg shrimp (ca. 3 cm in length and 1 ± 0.4 g in weight), collected from five regions in Taiwan, were analyzed using the World Organization of Animal Health (OIE) certified IQ2000^TM^ WSSV detection system, and the results showed that 6 out of 20 samples were positive (Figure 5A). According to the manufacturer’s instructions, the amount of WSSV in the six positive samples was greater than 2000 copies.

Next, the same 20 shrimp samples were examined using our direct PCR-based CE platform. Approximately one millimeter of pereiopod was removed from each shrimp and transferred to a lysis buffer for heat treatment. Subsequently, direct PCR was performed with 2.5 μL of crude extract, followed by CE analysis. The results also showed that 6 of the 20 samples were positive, and the positive samples were completely consistent with the results of the IQ2000^TM^ WSSV detection system (Figure 5B), indicating that our direct PCR-based CE platform is highly sensitive and specific compared with the IQ2000^TM^ WSSV detection system (Table 2). Moreover, comparing the time cost in each step of the two detection methods (Table 3), the direct PCR-based CE platform omits the steps of nucleic acid extraction and purification and simplifies the process of PCR product detection. The overall operation takes only 84 min, which is 117 min less than the IQ2000^TM^ WSSV kit.

### 3.4. Disease Detection in Individual Shrimp Post Larvae 

This platform was further used to detect potential carriers of EHP in post larvae shrimp. Ten whiteleg shrimp post larvae (ca. 0.5 cm in length and 3.3 ± 0.7 mg in weight) were collected from five different regions, and the single, whole post larvae underwent the same detection process, including lysis buffer-based heat treatment (30 μL mg^−1^), direct PCR for EHP sequence amplification, and CE analysis. The results showed that 4 out of 10 samples were positive for EHP; these four were from two aquaculture ponds (Figure 6).

## 4. Discussion

In this study, direct PCR combined with a CE platform was proposed as a solution for shrimp disease detection in on-site aquaculture. Our results confirmed that direct PCR-based CE platforms are characterized by simple operation, less time consumption, high specificity, and high sensitivity, making them suitable for post larvae shrimp examination and routine disease monitoring in the field.

Direct PCR has the advantage of omitting the nucleic acid extraction step, in addition to reducing the time needed and avoiding possible on-site contamination [27]. However, the presence of PCR-interfering substances in tissues may affect the sensitivity of the assay if these inhibitors are not removed through a purification process [28]. In the various shrimp tissues tested, we found interference problems in the hepatopancreas, stomach, and midgut tissues (Figure 1). Since the hepatopancreas is the target of various pathogens, such as *Vibrio* and EHP [32,33], overcoming this problem improved the practical value of direct PCR in shrimp disease detection.

Many reports have indicated that the use of protein engineering to enhance enzyme activity or the addition of PCR additives (e.g., betaine, dimethyl sulfoxide, and formamide) to the reaction buffer could ameliorate the interference caused by PCR inhibitors present in biological tissues [34,35,36]. Various manufacturers have their own formulations of direct PCR products, but these products do not necessarily target shrimp when they are developed. Therefore, we selected four brands of available products to test in direct PCR involving whiteleg shrimp hepatopancreas tissue, which contains a number of PCR inhibitors. The results showed that the interference inhibition effects of the four manufacturers’ formulas on shrimp hepatopancreas were indeed different (Figure 3A). Among them, DirectGO from Ten Giga Bio was more effective against PCR inhibitors present in shrimp tissue.

After PCR, correct analysis of the results is also a key point in developing a disease detection platform. Agarose-based electrophoresis analysis is time-consuming, difficult to implement on-site, and has poor resolution and sensitivity [24]. Therefore, we believe that small, portable CE is a better choice [37]. Our results showed that CE could increase the detection limit by at least four times under the same PCR conditions (Figure 2). However, a CE system with electrodynamic injection is more sensitive to salt ions in the samples [23]. Consequently, the selection of direct PCR reagents should also consider the salt concentration of the overall reaction solution to avoid interference from the CE injection that may cause the sensitivity to deteriorate.

Compared with the OIE-certified IQ2000^TM^ WSSV detection system, the direct PCR-based CE platform also had similar levels of detection sensitivity (Table 4). In order to effectively reduce PCR inhibitors in the reaction, the amount of sample dissolved in the lysis buffer must be limited (30 μL mg^−1^). Since the IQ2000^TM^ WSSV detection system includes nucleic acid extraction and purification, it can accommodate more tissue samples, which has made it more advantageous in detection sensitivity [14,38,39]. However, the direct PCR-based CE platform has obvious time-saving advantages in the two stages of sample preparation and result analysis [40,41]. The detection time of one sample can be shortened by approximately 117 min (Table 3) while maintaining the same detection sensitivity, which is already sufficient for disease detection. Thus, in the detection of 20 clinical samples, the results were completely consistent with the IQ2000^TM^ WSSV kit (Figure 5).

The IQ2000^TM^ WSSV system adopts nested PCR technology, which reduces false positives and increases the detection limit through two-stage amplification of two sets of primers [14,38,39]. In direct PCR-based CE, although only one set of primers is used for PCR, the high resolution of CE for fragment length allows it to clearly distinguish DNA products of correct length from non-specific amplified fragments [25]. In this way, the rate of false positives is reduced, and the PCR can be amplified with more cycles to enhance the signal. According to our results, the detection limit for WSD, HPM, and AHPND reached the level of 25 copies/PCR-reaction. Even in the hepatopancreas, a sample tissue with severe interference, the detection limit reached the level of 100 copies/PCR-reaction (Figure 4 and Table 4). Such a detection limit is equivalent to other commercially available disease detection kits and meets the requirements of clinical diagnosis. In the future, this system can also be added to IQ2000^TM^-competitive PCR design to expand the options for quantitative analysis [14,38,39].

With global climate change, the shrimp farming industry faces an increasing risk of infectious disease outbreaks. Although not yet classified as a serious shrimp infectious disease by the OIE, according to many reports, EHP has begun to cause significant economic losses to global shrimp aquaculture. Since there is currently no effective treatment for EHP, public health measures, such as strict quarantine, are still considered effective ways to block the spread of EHP [42]. Our data demonstrated that the use of a direct PCR-based CE platform enabled rapid and sensitive detection of EHP-infected post larvae (Figure 6). More importantly, the detection platform we proposed is easily operated on-site, especially for quarantine work before the shrimp post larvae enter the aquaculture pond or for the regular health management. Through more convenient infectious disease detection, quarantine and epidemic prevention work can be performed more easily to reduce the spread of deadly infectious diseases and protect the global shrimp aquaculture industry.

## Figures and Tables

**Figure 1 life-12-00276-f001:**
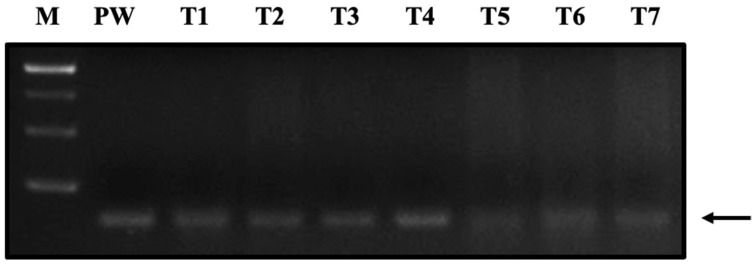
Evaluation of the interference of different tissues of juvenile shrimp on the amplification of WSSV pathogenic genes. A DirectGO direct PCR kit was the tested reagent. A 140 bp PCR amplification product of the VP28 gene was generated using the WSSV primer pair and 250 copies of standard DNA plasmid (pUC57-VP28). M: DNA marker; PW: ddH_2_O; T1: pereiopods; T2: pleopods; T3: gill; T4: hemolymph; T5: hepatopancreas; T6: stomach; and T7: midgut.

**Figure 2 life-12-00276-f002:**
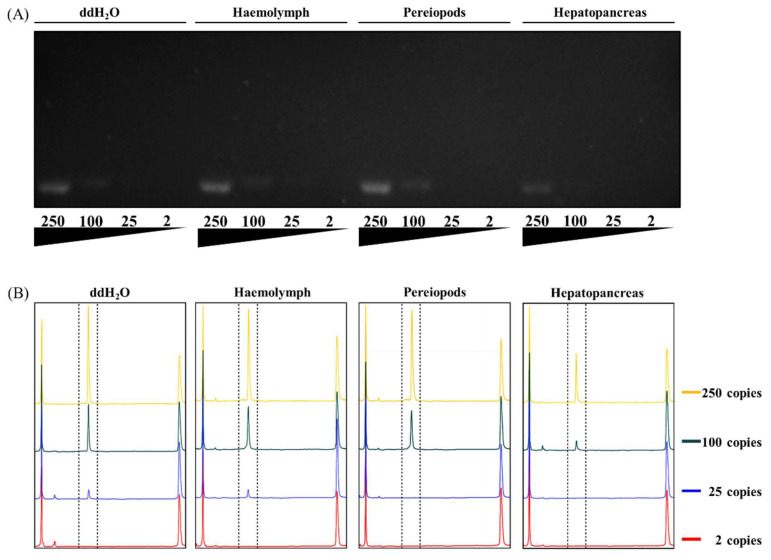
Detection limit comparison in the presence of hemolymph, pereiopod, or hepatopancreatic lysate. Direct PCR products were analyzed by (**A**) agarose gel electrophoresis and (**B**) capillary electrophoresis. The 140 bp PCR amplification product of the VP28 gene was generated using the WSSV primer pair and 2–250 copies of standard DNA plasmid.

**Figure 3 life-12-00276-f003:**
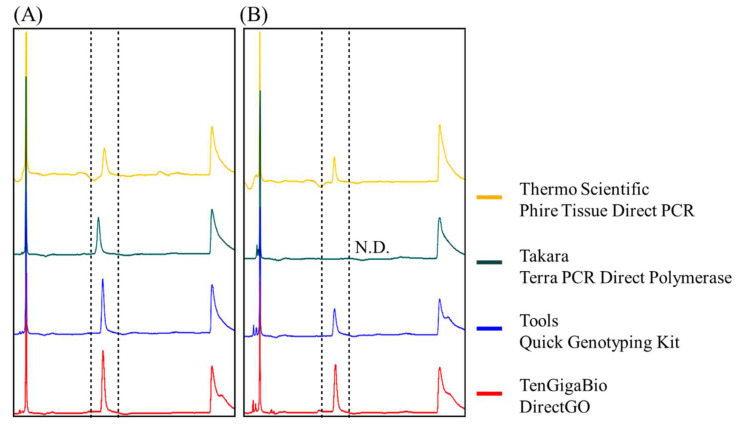
Comparison of the effect of four commercially available direct PCR kits on amplifying the VP28 gene in the presence of interfering hepatopancreatic lysates. (**A**) Control group: absence of hepatopancreatic lysate. (**B**) Analysis of direct PCR products in the presence of hepatopancreatic lysates. The PCR template was 250 copies of standard DNA plasmid. The number in the figure represents the peak area of the signal. N.D. means no detection.

**Figure 4 life-12-00276-f004:**
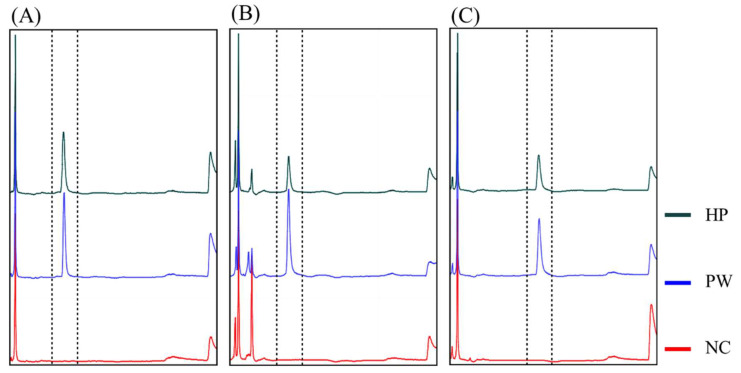
Detection of WSD, AHPND, and HPM pathogens using direct a PCR-based capillary electrophoresis platform. (**A**) WSD, (**B**) HPM, and (**C**) AHPND. HP: analysis of direct PCR products in the presence of hepatopancreatic lysates; PW: control group, absence of hepatopancreatic lysate; NC: negative control, absence of hepatopancreatic lysate and DNA polymerase. The PCR template was 250 copies of standard DNA plasmid.

**Figure 5 life-12-00276-f005:**
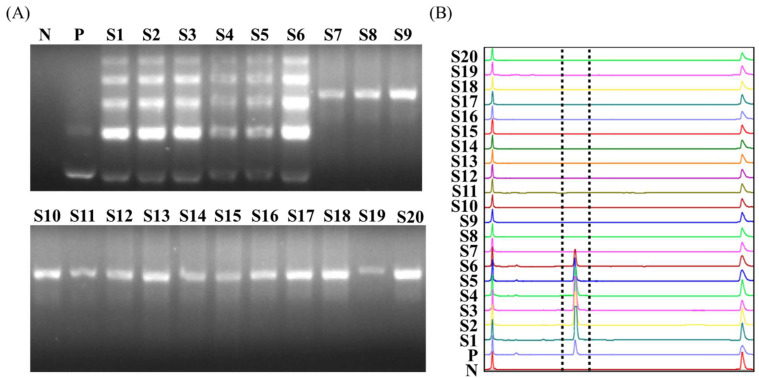
Comparison of detection sensitivity in clinical juvenile shrimp samples using two disease detection systems: (**A**) the IQ2000^TM^ WSSV detection system (gold standard), and (**B**) direct PCR-based capillary electrophoresis. N: negative control, reaction absence of DNA polymerase; P: positive control, replacement of shrimp lysate with standard plasmids in the reaction.

**Figure 6 life-12-00276-f006:**
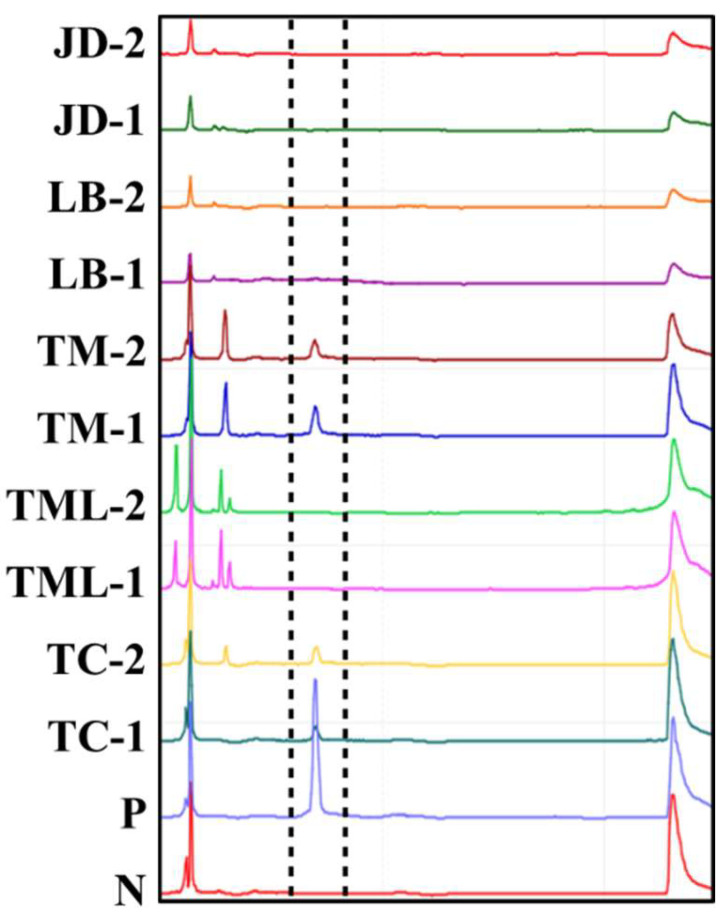
Pathogen detection of EHP in single post larva shrimp samples using direct PCR-based capillary electrophoresis. N: negative control, reaction absence of DNA polymerase; P: positive control, replacement of shrimp lysate with standard plasmids in the reaction. TC: Toucheng; TML: Taimali; TM: Taimu; LB: Linbian; and JG: Jiadong.

**Table 1 life-12-00276-t001:** The primers used in this study.

Disease	Target	Accession No.	Primer Name	DNA Sequence (5′-3′)	Anneal Temp.	Product Size	Reference
AHPND	Pir toxin gene	JALL01000066	AP4-F2	TTGAGAATACGGGACGTGGG	55 °C	230 bp	[29]
			AP4-R2	GTTAGTCATGTGAGCACCTTC			
HPM	spore wall protein gene	KX258197	SWP_2F	TTGGCGGCACAATTCTCAAACA	64 °C	148 bp	[30]
			SWP_2R	GCTGTTTGTCTCCAACTGTATTTGA			
WSD	VP28 envelope protein gene	AY249442	VP28-140Fw	AGGTGTGGAACAACACATCAAG	58 °C	140 bp	[31]
			VP28-140Rv	TGCCAACTTCATCCTCATCA			

**Table 2 life-12-00276-t002:** Diagnostic performance of direct PCR-based capillary electrophoresis for detecting the WSSV gene in clinical samples of shrimp with WSSV.

		IQ2000 (Gold Standard)				
		Positive	Negative	Sensitivity (%)	Specificity (%)	Agreement (%)
DirectGO	Positive	6 (TP)	0 (FP)	100	100	100
	Negative	0 (FN)	14 (TN)			

**Table 3 life-12-00276-t003:** Comparison of the time cost for WSSV disease detection by the IQ2000^TM^ WSSV detection system and direct PCR-based capillary electrophoresis.

Procedure	IQ2000	DirectGO
Pre-PCR		
Sample treatment	~50 min	~20 min
PCR Premix	~5 min	~5 min
PCR		
1st PCR	21 min	50 min 20 s
PCR mature prepare	~5 min	-
2nd PCR	30 min 10 s	-
Post-PCR		
Make agarose gel	30 min	-
Electrophoresis	35 min	~3 min
Gel stain and destain	20 min	-
Data assay	5 min	1 min
Total time	>201 min	>86 min

**Table 4 life-12-00276-t004:** Comparison of the PCR detection profile for WSSV disease detection with the IQ2000^TM^ WSSV detection system and direct PCR-based capillary electrophoresis.

Item	Direct PCR-Based CE	IQ2000
Tissue weight	~3 mg	~20 mg
Total sample volume after extraction	90 uL	200 uL
Sample added per PCR reaction	2.5 uL	2 uL
Equivalent tissue weight per PCR reaction	~83 ug	~200 ug
Limitation of detection per PCR reaction (LOD)	25–100 copies	20 copies

## Data Availability

The data presented in this study are available on request from the corresponding author.

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
