# Peer review of "Combining Direct PCR Technology and Capillary Electrophoresis for an Easy-to-Operate and Highly Sensitive Infectious Disease Detection System for Shrimp"

_life, 2022, doi:10.3390/life12020276_

Round 1

Reviewer 1 Report

The paper describes interesting and feasible methodology to detect various pathogens in shrimp farming. My minor comments were provided below:

L58/54 - Please expand this paragraph and describe background for the capillary methods.

L65/68 - Please formulate a clear aim of the study.

Subchapter 2.1 Materials – Information on the equipment, kits, etc. should be included in the next subchapters. There is no need to list them all in one chapter.

L112 – Details included in 2.1 should be provided e.g. here for portable capillary electrophoresis.

L118 – Please specify OIE recommendations.

L119 and 121– Delete spaces between unit and value. Update also other instances.

L143/147 – This part of the results should be moved to M&M section.

Author Response

Responses to Reviewer’s comments

We find the comments and suggestions made by the reviewers very genuine and helpful, and highly appreciate them for such constructive comments. Detail responses to each comment are as follows:

Reviewer #1: The paper describes interesting and feasible methodology to detect various pathogens in shrimp farming. My minor comments were provided below:

  1. L58/54 - Please expand this paragraph and describe background for the capillary methods.

Response: We have modified the third paragraph (Line 61-67) of Introduction section with additional information.

Another example is capillary electrophoresis (CE), which reduces post-PCR analysis time [23]. CE has been applied to various high-throughput nucleic acid analysis applications, including genotyping and pathogen detection, due to its high sensitivity, superior accura-cy, speedy, and automation [23-25]. The recent development of lightweight and portable CE machine makes this powerful tool becoming an excellent apparatus for on-site nucleic acid detection [26].

Reference

  1. Righetti, P.G.; Gelfi, C. Capillary electrophoresis of DNA in the 20–500 bp range: Recent developments. Biochem. Biophys. Methods 1999, 41, 75-90.
  2. Slater, G.W.; Desruisseaux, C.; Hubert, S.J.; Mercier, J.F.; Labrie, J.; Boileau, J.; Tessier, F.; Pépin, M. P. Theory of DNA electrophoresis: A look at some current challenges. Electrophoresis 2000, 21, 3873-
  3. Liu, Y.; Huang, X.; Ren, J. Recent advances in chemiluminescence detection coupled with capillary electrophoresis and microchip capillary electrophoresis. Electrophoresis 2016, 37, 2-
  4. Kerékgyártó, M.; Kerekes, T.; Tsai, E.; Amirkhanian, V.D.; Guttman, A. Light-emitting diode induced fluorescence (LED-IF) detection design for a pen-shaped cartridge based single capillary electrophoresis system. Electrophoresis 2012, 33, 2752-

  1. L65/68 - Please formulate a clear aim of the study.

Response: Your kindly suggestion was followed. We have modified the final paragraph (Line 70-78) of Introduction section.

Direct PCR technology can omit the nucleic acid purification step and directly analyze shrimp tissue samples [27]. But the PCR reaction may be interfered by PCR inhibitors from tissues thus reduce the feasibility to use for diagnostics [28]. In this work, we careful-ly compared different direct PCR systems and found the best system for shrimp tissue analysis. Using portable CE to perform post-PCR analysis, the detection sensitivity, specificity, and efficiency could be further improved [24]. By comparing with an OIE-certified system for disease detection of white shrimp in aquaculture ponds from different regions, we have verified that this technology platform (i.e., direct PCR based CE) is suitable for on-site disease surveillance in aquaculture.

  1. Subchapter 2.1 Materials – Information on the equipment, kits, etc. should be included in the next subchapters. There is no need to list them all in one chapter.

Response: Your kindly suggestion was followed. We have added subchapter 2.2 Reagents and equipment to describe equipment and reagents.

  1. L112 – Details included in 2.1 should be provided e.g. here for portable capillary electrophoresis.

Response: Thanks for the reviewer's suggestion. Its description has been covered in Line 91-93.

  1. L118 – Please specify OIE recommendations.

Response: Thank to the reviewer to point it out. We have cited a paper (Ref. 14) here to specify OIE recommendations.

  1. L119 and 121– Delete spaces between unit and value. Update also other instances.

Response: Your kindly suggestion was followed.

  1. L143/147 – This part of the results should be moved to M&M section.

Response: Thanks for the reviewer's suggestion. We have revised the sentence in to more focus on the experimental conditions and results (Please see highlight in manuscript Line 156-161). In addition, the experimental steps of this part have been detailed in subchapter 2.3 Direct-PCR for the Detection of Shrimp Disease.

Reviewer 2 Report

See attachment

Author Response

Responses to Reviewer’s comments

We find the comments and suggestions made by the reviewers very genuine and helpful, and highly appreciate them for such constructive comments. Detail responses to each comment are as follows:

Reviewer #2: This manuscript by Lin et al., titled “Combining Direct PCR Technology and Capillary Electrophoresis to Implement Easy-to-Operate and Highly Sensitive Shrimp Infectious Disease Detection” explores how a sensitive, easy-to-operate PCR-based method combined with capillary electrophoresis can be used for field disease detection in shrimp aquaculture. Overall, the study is well designed and executed, with the findings of use and interest to shrimp aquaculture and indeed aquaculture in general. That said, there are a few things that need to be revised to put the paper in a better shape. Details are outlined below:

  1. I think a more succinct title could be something like “Combining direct PCR technology and capillary electrophoresis for an easy-to-operate and highly sensitive infectious disease detection system for shrimp”

Response: Your kindly suggestion was followed.    

  1. Lines 20-21 “Disease outbreaks are preventable due to the specificity, rapidity, and sensitivity of the molecular diagnostic methods used to detect asymptomatic infected shrimp” this statement is not entirely true because detection of a disease does not necessarily lead to its prevention, so please revise this.

Response: Thanks for the reviewer's suggestion. We have revised the sentence in the Abstract (Please see highlight in manuscript Line 20-21).

  1. Lines 24-25, “Combined with capillary electrophoresis analysis, pathogen detection limit could reach 25 copies/reaction of tissue.” This is a hanging sentence, not properly linked with preceding one, so please check and rewrite.

Response: Thank to the reviewer’s advice. We have changed the sentence to “In addition, portable capillary electrophoresis was applied to improve sensitivity and specificity, resulting in a pathogen detection limit of 25 copies/PCR-reaction.“

  1. Line 30, “single shrimp post-larvae.” Please change to "single post-larvae shrimp"

Response: Your kindly suggestion was followed.

  1. Lines 41-43, “Major shrimp infectious disease outbreaks on a global scale occur every few years, including white spot syndrome virus (WSSV), acute hepatopancreatic necrosis disease (AHPND), and Enterocytozoon hepatopenaei (EHP)...” This statement seems to suggest that WSSV and EHP are diseases. These are the agents/pathogens that cause the disease; hence, this statement should be revised succinctly.

Response: Thanks for the reviewer's suggestion. We have revised disease names consistently throughout the mauescript.

  1. Line 44, “annual economic loss...”, Is this annual economic loss global or regional, e.g., Asia? please state clearly.

Response: We apologize for the confusion. We have replaced Ref. 7 with the below paper to match global data.

Referecnce

Oakey, J.; Smith, C.; Underwood, D.; Afsharnasab, M.; Alday-Sanz, V.; Dhar, A.; Sivakumar, S.; Sahul Hameed, A. S.; Beattie, K.; Crook, A. Global distribution of white spot syndrome virus genotypes determined using a novel genotyping assay. Arch. Virol. 2019, 164, 2061-2082.

  1. Line 46, “demonstrated during COVID-19 pandemic”, Please revise to "during the ongoing COVID-19 pandemic"

Response: Your kindly suggestion was followed.

  1. Line 56, “the consumption of time and money for detection...”, Please delete “consumption of” and insert "needed" after money.

Response: Your kindly suggestion was followed.

  1. Line 64, “causes sample...”, please replace “causes” with "results in"

Response: Your kindly suggestion was followed.

  1. Line 66, “analyze the shrimp tissue samples. Coupled with capillary analysis, the method...” Please delete “the” before “shrimp”. Th next sentence should begin like "When coupled with capillary analysis, this method..."

Response: Your kindly suggestion was followed.

  1. Line 68, please insert “and therefore” between “detection suitable”

Response: Your kindly suggestion was followed.

  1. Line 71, “... were provided by TenGigaBio...” Were these reagents bought or provided for free?, please state clearly.

Response: Thanks for the reviewer's suggestion. We have modified the sentence to clarify the state in the Materials and Methods.

  1. Lines 90-93, “The organs of the juvenile shrimp were collected using dissection tools for their muscles, pereiopods, pleopods, gills, hemolymph, hepatopancreas, stomach, and midgut. Subsequently, the weights of these tissues were measured, and they were stored at −80 °C.” These two sentences could be linked succinctly as follows: “The organs (i.e., muscles, pereiopods, pleopods, gills, hemolymph, hepatopancreas, stomach, and midgut) of the juvenile shrimp were collected using dissection tools before being weighed and stored at −80 °C.”

Response: Your kindly suggestion was followed.

  1. Line 118, “the OIE recommendations.” Please insert the relevant reference here.

Response: Thank to the reviewer to point it out. We have cited the correct reference (Ref. 14).

  1. Line 150 “signal was affected”, What is meant by this expression? please clarify

Response: Thanks for the reviewer's suggestion. We have rewritten the sentences with more specific words in Results.

  1. Line 166, “limitations of the operating site”, What is meant by this expression? please clarify.

Response: We have correct the phrase into “more complex manual manipulation”.

  1. Lines 181-182, “Next, we used the same infusion to perform direct PCR on the hepatopancreas tissue, with strong PCR reaction interference.” Please rephrase this sentence to clarify the meaning.

Response: We have rewritten the sentence into “Next, we used the same infusion to perform direct PCR on the hepatopancreas tissue to determine whether nucleic acid amplification efficiency was affected by interference from shrimp tissue extracts”.

  1. Line 195, “potent peak”, What is meant by this term? please clarify

Response: We apologize for the confusion. We have correct the words into “specific and strong peak”.

  1. Line 254, “time consumption”, Please replace “consumption” with "needed"

Response: Your kindly suggestion was followed.

  1. Lines 259-260, “overcoming this problem has promoted the improved practical value of direct PCR in shrimp disease detection.”, Was this problem overcome or how would you overcome this problem? Please delete “has promoted” and insert “the” after “improved”

Response: We have already mentioned in the next paragraph (Line 280-285) that these PCR inhibitors in the hepatopancreas of shrimp can be overcome using the DirectGO kit (TenGigaBio). In addition, we have modified the sentence as suggested.

  1. Line 268, “tissue with a number”, Please replace “with a” with "which contains a..."

Response: Your kindly suggestion was followed.

  1. Line 268-270, “The results show that the interference inhibition effects of various manufacturers’ formulas on the shrimp hepatopancreas were indeed different (Figure 3A).” Please revise this statement succinctly to clarify the meaning.

Response: We have modifed the sentence into “The results show that the interference inhibition effects of the four manufacturers’ formulas on the shrimp hepatopancreas were indeed different”.

  1. Line 271, “inhibitors derived from shrimp tissue.”, Please replace “derived from” with “present in”

Response: Your kindly suggestion was followed.

  1. Lines 274-275, “analysis takes a long time, is difficult”, Please replace “takes a long time” with "is time consuming" and delete “is”

Response: Your kindly suggestion was followed.

  1. Lines 286-287, “PCR inhibitor in the reaction, the amount of sample collection must be limited. Since the IQ2000TM WSSV detection system has undergone...”, Please add “s” to “inhibitor”. The expression “the amount of sample collection must be limited.” is quite vague, please clarify. Also replace the expression “has undergone” with "includes"

Response: We have followed the reviewer's suggestions and correct the phrase into” the amount of sample dissolved in the lysis buffer must be limited (30 μL mg-1).”

  1. Line 293, “20 real samples”, What is meant by this? do we have unreal samples? please clarify.

Response: Thanks for the reviewer's suggestion. We have correct the words into “20 clinical samples”.

  1. Lines 302-303, “our actual verification”, Please replace with a better expression

Response: Thanks for the reviewer's suggestion. We have correct the words into “According to our results”.

  1. Line 306, “commercially available level”, Please clarify this expression.

Response: Thanks for the reviewer's suggestion. We have correct the words into “other commercially available disease detection kits”.

  1. Lines 306-307, “meets the demands of clinical diagnosis” Is it the requirements or demands of clinical diagnosis? Please check and clarify.

Response: Thank to the reviewer to point it out. We have correct the words.

  1. Line 311, “the high-density shrimp aquaculture industry”, I’m not sure if this is the appropriate term to use in this context, so please check and revise.

Response: Thanks for the reviewer's suggestion. We have correct the words into “shrimp farming industry”.

  1. Line 320, “or the daily disease monitoring.”, This clause is not well linked with the main clause, so please check and revise.

Response: Thank to the reviewer to point it out. We have correct the words into “the regular health management”.

Reviewer 3 Report

graphical abstract is unsufficient

Author Response

Responses to Reviewer’s comments

We find the comments and suggestions made by the reviewers very genuine and helpful, and highly appreciate them for such constructive comments. Detail responses to each comment are as follows:

Reviewer #3:

  1. graphical abstract is unsufficient

Response: Thanks for the reviewer's suggestion. We have modified the graphic summary and added some important information.
